



# The future ozone trends in changing climate simulated with SOCOLv4

Arseniy Karagodin-Doyennel[1,2], Eugene Rozanov[1,2,3], Timofei Sukhodolov[1,3], Tatiana Egorova[1], Jan Sedlacek[1], and Thomas Peter[2]

[1]Physikalisch-Meteorologisches Observatorium Davos/World Radiation Center (PMOD/WRC), Davos, Switzerland
[2]Institute for Atmospheric and Climate Science (IAC), ETH, Zurich, Switzerland
[3]Saint Petersburg State University, Saint Petersburg, Russia

**Correspondence:** Arseniy Karagodin-Doyennel (darseni@student.ethz.ch)

**Abstract.** This study evaluates the future evolution of atmospheric ozone between 2015 and 2099 simulated with the Earth System Model (ESM) SOCOLv4. Simulations have been performed based on two potential Shared Socioeconomic Pathways (SSP): the "middle-of-the-road" (SSP2-4.5) and "fossil-fueled" (SSP5-8.5) scenarios. In both scenarios, the model projects a decline in tropospheric ozone in the future that starts in the 2030s in SSP2-4.5 and after the 2060s in SSP5-8.5 due to a decrease in concentrations of ozone precursors like $NO_x$ and CO. The results also suggest a very likely ozone increase in the mesosphere, upper and middle stratosphere, as well as in the lower stratosphere at high latitudes. Under SSP5-8.5, the ozone increase in the stratosphere is higher because of stronger cooling (> 1°K/decade) induced by the greenhouse gases (GHG), which slows the catalytic ozone destruction cycles. In contrast, in the tropical lower stratosphere ozone concentrations decrease in both experiments and increase over the middle and high latitudes of both hemispheres due to the intensification of meridional transport, which is stronger in SSP5-8.5. No evidence was found of a decline in ozone levels in the lower stratosphere at mid-latitudes. In both future scenarios, the total column ozone is expected to be distinctly higher than present in mid-to-high latitudes and might be lower in the tropics, which causes a decrease/increase in the surface level of UV radiation. The results of SOCOLv4 suggest that the stratospheric ozone evolution throughout the 21[st] century is strongly governed not only by a decline in halogen concentration but also by future GHGs forcing. In addition, the tropospheric ozone column changes, mainly due to the changes in anthropogenic emissions of ozone precursors, also have a strong impact on the total column. Therefore, even though the anthropogenic halogen loading problem has been brought under control to date, the sign of future ozone column changes, globally and regionally, is still unclear and largely depends on diverse future human activities. The results of this work are, thus, relevant for developing future strategies for socioeconomic pathways.

## 1 Introduction

The stratospheric ozone layer is an essential element of the Earth's atmosphere. It shields the ecosystem from dangerous ultraviolet radiation, shapes the vertical temperature profiles, and thus affects the general circulation of the atmosphere. The tropospheric ozone is one of the most potent greenhouse gases (e.g., Allan et al., 2021), contributing to the rise in near-surface temperature, as well as a toxic air pollutant harmful to human health and vegetation. Thus, ozone contributes to climate change





and human, agriculture, and ecosystem development (e.g., Barnes et al., 2019).

A serious challenge for humanity is the consequences of stratospheric ozone depletion caused by man-made halogenated ozone-depleting substances (hODS). This prompted the nations to ratify the Montreal Protocol in 1987, an international treaty to phase out hODSs. The Montreal Protocol and its Amendments and Adjustments (MPA) allows the ozone layer to recover from the hODS effect. Various studies show that the total ozone column decrease has been reversed at most latitudes, which is

attributed to the decline in hODS concentrations, highlighting the success of the MPA in protecting the ozone layer (Newchurch et al., 2003; Solomon et al., 2016; WMO, 2018; McKenzie et al., 2019). Different projections of the future ozone layer evolution during the 21$^{st}$ century suggest that the decrease in hODS facilitates ozone recovery in the stratosphere (Banerjee et al., 2016). Ozone abundances are expected to return to the pre-1960 level in most atmospheric layers by the mid-to-late century, except in the lower stratosphere (Eyring et al., 2007; Austin et al., 2010; Dhomse et al., 2018). Thus, it is believed that de-

clining hODS will gradually lose their leading role in determining the evolution of the ozone layer throughout the 21$^{st}$ century (Newman, 2018).

Studies claim that greenhouse gases (GHGs), such as carbon dioxide ($CO_2$), methane ($CH_4$), and nitrous oxide ($N_2O$) will largely control ozone changes in the 21$^{st}$ century. $CO_2$ facilitates the stratospheric ozone enhancement due to direct radiative

cooling of the stratosphere, slowing down the gas-phase ozone destruction rate (Randeniya et al., 2002; Stolarski et al., 2015). Therefore, for some parts of the stratosphere, even a "super-recovery" is expected, i.e., ozone levels well above pre-1980s level (Eyring et al., 2007; Meul et al., 2016).

Whilst $N_2O$ is mainly inert in the troposphere, the growth of its concentration will hamper the increase of the stratospheric

ozone in the future (Ravishankara et al., 2009; Chipperfield, 2009; Revell et al., 2012, 2015; Stolarski et al., 2015), due to the increased production of nitrogen oxides ($NO_x$= NO + $NO_2$), which catalytically destroy ozone (Crutzen, 1970). Yet, the GHG-related cooling of the stratosphere may reduce the efficiency of catalytic cycles involving $NO_x$. This is due to the fact that more $NO_x$ is converted to inactive $N_2$, i.e., the $N_2O$ contribution to ozone destruction can be somewhat lowered (Revell et al., 2015).

$CH_4$ plays an ambivalent role in ozone change as it may have both negative and positive effects on ozone. The negative effect of increased $CH_4$ on stratospheric ozone is that it increases the efficiency of the hydroxyl oxide ($HO_x$) catalytic cycle of ozone destruction since $CH_4$ is the main source of $H_2O$ in the middle atmosphere (Bates and Nicolet, 1950). However, it should be noted that additional $HO_x$ and $NO_x$ radicals would also partly compensate for the negative effects of each other in the stratosphere through the production of reservoir species $HNO_3$ ($OH + NO_2 + M \rightarrow HNO_3 + M$). $CH_4$ also has a positive effect

on ozone, as it causes an additional chlorine deactivation ($CH_4 + Cl \rightarrow CH_3 + HCl$) throughout the stratosphere (Hitchman and Brasseur, 1988) and promotes an increase in tropospheric ozone by being a source of CO (Brasseur and Solomon, 2005; Morgenstern et al., 2013), that is a precursor for ozone formation in the lower atmosphere.





The future evolution of tropospheric ozone will be strongly driven by the changes in CO and $NO_x$, leading to large dif-
ferences in projections of tropospheric ozone for distinct climate scenarios (Revell et al., 2015b; Archibald et al., 2020). In
addition, the projections indicate that the future ozone changes in the troposphere are more non-linear than in the stratosphere
(Revell et al., 2015b).

Most chemistry-climate models (CCMs) project that the ozone layer will continue to thin in the tropical lower stratosphere
throughout the 21st century (Zubov et al., 2013; Banerjee et al., 2016; Keeble et al., 2021). The speed of this thinning depends
on the climate scenario for GHGs (Dhomse et al., 2018; Shang et al., 2021). GHG-induced temperature changes in the lower
atmosphere strengthen the meridional transport via the shallow branch of Brewer-Dobson circulation (BDC) due to an increase
in temperature gradient between tropical and mid-latitudes. This raises the tropopause, alters the wave propagation and dissi-
pation, and extends the subtropical transport barriers upward (Zubov et al., 2013; Butchart, 2014; Chiodo et al., 2018; Abalos
and de la Cámara, 2020). The faster atmospheric upwelling decreases the ozone production in the ascending air parcel (Aval-
lone and Prather, 1996). The intensified transport also increases the stratosphere-troposphere exchange with more ozone-poor
tropospheric air being transported to the lower stratosphere (WMO, 2018). Models also exhibit significant differences in the
magnitude of the simulated GHG-induced acceleration of the BDC (Morgenstern et al., 2018).

Projections of the ozone layer and, hence, of the future surface UV levels strongly depend on the GHGs scenarios applied,
especially by the end of the 21st century (Butler et al., 2016). Current IPCC CMIP6 activities (Eyring et al., 2016) have
developed GHG emission scenarios based on Shared Socioeconomic Pathways (SSP), which take economic, demographic, and
technological perspectives into account (O'Neill et al., 2016; O'Neill et al., 2017; Riahi et al., 2017; Zhang et al., 2019). There-
fore, an important task is to examine the sensitivity of the ozone evolution to different contemporary GHG-scenarios applied.
In Shang et al. (2021), it was done as an intercomparison of three CMIP6 models under several SSP scenarios (SSP1-2.6,
SSP2-4.5, SSP3-7.0, SSP5-8.5). The general ozone increase in the global stratosphere has been demonstrated for all employed
scenarios. Also, all GHG-scenarios contribute positively to closing the Antarctic ozone hole. However, the projected changes
in the tropical stratospheric ozone column are shown to scale non-linearly with the growth of social development, i.e., with
incrementing GHGs emissions.In addition, Shang et al. (2021) showed that due to the decline in lower stratospheric ozone,
the tropical ozone column is expected to be largely determined by tropospheric ozone abundance, which might be higher, if the
SSP5-8.5 scenario plays out. By analyzing simulations with CMIP6 models under various SSP scenarios, Keeble et al. (2021)
showed that under SSP5-8.5, the total ozone column is expected to be 10 DU higher than its 1960 level. On the contrary, total
tropical column ozone is not predicted to return to 1960 levels in most of the SSP scenarios, due to either tropospheric or lower
stratospheric ozone decrease (Keeble et al., 2021). Revell et al. (2022) showed the importance of simulating stratospheric
ozone accurately for Southern Hemisphere climate change projections, in particular of wind, by comparing CMIP6 model
simulations performed with and without interactive chemistry under moderate (SSP2-4.5) and high (SSP5-8.5) SSP scenarios.
Their results demonstrate inconsistency between simulations with and without interactive chemistry, showing differences in
temperature and westerly wind patterns in the Southern Hemisphere driven by differences in Antarctic springtime ozone. This





underscores the importance of accurately modeling ozone changes for future climate projections.


The quantitative analysis of ozone changes can be promoted by state-of-the-art regression models, utilizing a complex and robust statistical approach to diagnose ozone trends. In fact, trends might vary significantly between regression models since they depend strongly on the type of model, time series length, and analysis applied (Hood and Soukharev, 2018). Using an advanced type of regression modeling, Dynamic Linear Modeling (DLM), to analyze space-borned ozone measurements, Ball

et al. (2018) provided evidence for an ongoing ozone decrease in the mid-latitude lower stratosphere despite the ozone recovery from the decline in hODSs. CCMs still incapable of fully reproducing these trends, yet exhibiting some marginally significant signs of ozone decline, which are slightly comparable with observations (Karagodin-Doyennel et al., 2022). The model projections also show no evidence of future lower stratospheric ozone decrease at mid-latitudes, whereas they do project the ozone decline in the tropics (Zubov et al., 2013; Banerjee et al., 2016). This questions the ability to accurately simulate future ozone

evolution in mid-latitudes, including the most densely populated regions. In essence, asserting the statistical significance and robustness of ozone trends in the lower stratosphere is not straightforward due to large uncertainties induced by natural variability (WMO, 2018; Karagodin-Doyennel et al., 2022).

In addition, there are newly detected chemical drivers, whose absence in CCMs might further increase uncertainties in

ozone projections. Laube et al. (2014) describes two recently detected hODSs (CFC-113a and HCFC-133a), which reveal rising emission rates and are not regulated by MPA, but may have a significant impact on stratospheric ozone. Meanwhile, it was suggested that very short-lived substances (VSLSs), not regulated by MPAs, such as $CH_2Cl_2$, the atmospheric concentration of which has increased in the last years, could have significant implications for the ozone layer and climate (Hossaini et al., 2015, 2017). They indicate that VSLSs lead to a 2-6% more ozone depletion in near-global total column ozone, which

is especially due to upper tropospheric/lower stratospheric ozone depletion. (Hu et al., 2016) reported that in some regions carbon tetrachloride ($CCl_4$) emissions, controlled by the MPA, exceed the expectations given by the MPA by about two orders of magnitude. Also of concern is the model assessment of the ozone response to the evolution of bromine-containing VSLSs (VSLS-Br) produced by biogenic ocean sources (Zubov et al., 2013), because their mixing ratios have been prescribed for the entire period in the previous CCMI campaign. It is deemed that the emissions of VSLS-Br hinge on the changing climatic

parameters, such as surface wind speed and the ocean temperature because of their contribution to the ocean-atmosphere interaction and biota life cycle. Using an idealized mode design, it was claimed that an additional 5 pptv of VSLS-Br might cause ∼10 years of lagging in the ozone recovery in southern high latitudes (Oman et al., 2016). In addition, a 10% growth of VSLS-Br influx to the atmosphere using fixed ocean water composition is found (Falk et al., 2017). However, to increase the modeled future ozone trend accuracy, it is necessary to proceed with this study using an Earth System Model (ESM) with

an interactive ocean module enabled that can simulate the plankton life cycle, as the ESM-calculated VSLS-Br emissions may differ from the estimates calculated with the model with the prescribed ocean (Falk et al., 2017). This study is also motivated to evaluate whether the inclusion of these species results in any noticeable effect on ozone evolution. Thus, it is essential to attribute all the above-mentioned factors to properly model the future atmospheric ozone evolution under different climate





scenarios.


In this study, we assess future atmospheric ozone evolution simulated with SOCOLv4 ESM for the period 2015-2099. To provide the estimates for ozone trends, we carried out two sets of simulations, where the prescribed future GHGs evolution and tropospheric ozone precursors follow either the SSP2-4.5 or SSP5-8.5 scenario, respectively. Changes are derived by employing the advanced dynamic linear modeling algorithm, which has proven itself as a flexible regression tool for quantifying highly variable ozone changes (Ball et al., 2018; Karagodin-Doyennel et al., 2022). Section 2 outlines the computational methods and experiment design. The results of this study are provided in Section 3 followed by the discussion and conclusions summarized in Section 4.

## 2 Computational methods

### 2.1 The SOCOLv4 ESM description

In this study, simulations were performed with the Earth System Model (ESM) SOCOLv4.0 (SOlar Climate Ozone Links, version 4) (hereinafter SOCOLv4). SOCOLv4 consists of the Max Planck Institute for Meteorology (MPI-M) ESM version 1.2 (MPI-ESM1.2) (Mauritsen et al., 2019), the chemical module MEZON (Rozanov et al., 1999; Egorova et al., 2003) and the size-resolving sulfate aerosol microphysical module AER (Weisenstein et al., 1997; Sheng et al., 2015; Feinberg et al., 2019). MPI-ESM1.2 contains the general circulation model MA-ECHAM6 (the Middle Atmosphere version of the European Centre/Hamburg Model, version 6) to compute atmospheric transport, physics, and radiation transfer; the Hamburg Ocean Carbon Cycle (HAMOCC); the Max-Planck-Institute for Meteorology Ocean Model (MPIOM), and Jena Scheme for Biosphere-Atmosphere Coupling in Hamburg (JSBACH). A chemical solver is based on the Newton–Raphson implicit iterative method (Ozolin, 1992; Stott and Harwood, 1993) that includes approximately 100 chemical compounds, 216 gas-phase, 72 photochemical, and 16 stratospheric heterogeneous reactions on polar stratospheric clouds particles and in aqueous sulfuric acid aerosols. The update for MEZON in SOCOLv4, compared to its previous version used in CCMI-1, also includes several newly discovered and unregulated hODSs as well as additional chlorine- and bromine-containing VSLSs uncontrolled by the MPA (see Sukhodolov et al. (2021)). The advection scheme of Lin and Rood (1996) operates the transport of chemical species. Photolysis rates are calculated using a lookup-table approach (Rozanov et al., 1999), including the effects of the solar irradiance variability. MA-ECHAM6, MEZON, and AER are interactively coupled, exchanging gas concentrations, sulfate aerosol properties, and meteorological fields.

SOCOLv4 is formulated on the T63 horizontal resolution, which corresponds to ∼1.9°x1.9° and uses 47 vertical levels in hybrid pressure coordinates between Earth's surface and 0.01 hPa (∼80 km). The 15-min time step is used in SOCOLv4 to calculate dynamic processes, while chemistry and radiation calculations are performed every 2 hours. SOCOLv4 reproduces well the distribution of atmospheric tracers, climatology, and variability of the temperature/circulation fields. Details of the SOCOLv4 model description and validation can be found in Sukhodolov et al. (2021).



## 2.2 Experiment design

Here, we analyze two types of transient simulations, spanning the 2015-2099 period, based on projections of GHGs emissions from the up-to-date climate scenarios under the Shared Socioeconomic Pathways (SSPs; Riahi et al., 2017). In our study, simulations are performed using two selected SSPs scenarios representing pathways of "middle-of-the-road" (SSP2-4.5) and "fossil-fueled" (SSP5-8.5) development (O'Neill et al., 2016; O'Neill et al., 2017; Riahi et al., 2017; Zhang et al., 2019). Under these scenarios, the surface temperature is expected to rise by about 3°C and 5°C at around 2100, respectively Zhao et al. (2020).

The SOCOLv4 simulations are conducted under standard conditions. This means that runs were initiated from MPI-ESM 1.2 restart files for 1970 and chemistry was initiated from SOCOLv3 runs (Revell et al., 2016). This experiment was carried out starting from the year 1949. In 1980, the experiment was divided into ensemble members, which were initialized with slightly changing initial conditions, namely with a small (about 0.1%) perturbation of the first-month $CO_2$ concentration. From 2015, all historical climate forcings (following the recommendations of CMIP6 (Eyring et al., 2016)) are branched to either SSP2-4.5 or SSP5-8.5 scenarios using projected GHGs concentrations. The future solar irradiance projection is provided by HEPPA/SOLARIS as it is also recommended for CMIP6 (Eyring et al., 2016). Each experiment consists of three ensemble members in order to properly address the internal model variability impact on ozone evolution and to assess the level of statistical significance of the obtained results. In this study, we analyze trends in the ensemble mean ozone time series as well as chemical drivers and temperature.

## 2.3 Dynamic linear modeling (DLM)

We employ DLM (Laine et al., 2014; Alsing, 2019) to quantify long-term ozone evolution. DLM is a stochastic model to explain the natural or anthropogenic variability in times series using explanatory/proxy variables. Its application for the historical ozone trends and a detailed description can be found in previous studies (Laine et al., 2014; Ball et al., 2018, 2019, 2020; Alsing, 2019; Karagodin-Doyennel et al., 2022).

In this study, the DLM setup includes time series of several statistically independent explanatory variables, attributing to the known ozone variability, which are commonly used for regression analysis of ozone time series (WMO, 2018). These include the projection of total solar irradiance (TSI, W m$^{-2}$) (Matthes et al., 2017), the El Niño–Southern Oscillation variability represented by ENSO's 3.4 index (ENSO, degree K), calculated from the sea surface temperature field; equatorial zonal winds at 30 and 50 hPa, which are two principal components of the Quasi-biennial oscillation variability (QBO30 and QBO50, m s$^{-1}$); a stratospheric aerosol optical depth (SAOD, dimensionless) is determined by the aerosol extinction at 300-500 nm band, and the Arctic and Antarctic Oscillation indexes (AO and AAO, hPa), calculated from the geopotential height fields at 1000 and 700 mb pressure levels. These proxies are prepared for each ensemble member of both experiments (except for TSI, which is the same for all simulations) (see Appendix Figure A1). These proxies are orthogonal, have admissible covariance, and can





be used in the regression analysis (see Appendix Figure A2). The DLM also accounts for a first-order autoregressive (AR1) process  (Tiao et al., 1990). In addition, DLM estimates 6- and 12-month harmonics for the seasonal cycle.

The advantage of DLM against conventional multiple linear regression is that DLM accounts for the level of trend nonlinearity as a free parameter, allowing the trend to evolve over time. This nonlinearity parameter is inferred from the data along
with the trend term, seasonal cycle, proxy amplitudes, and the AR1 process  (Laine et al., 2014). In principle, this makes the DLM method more accurate for capturing the ozone variability, especially for the after-turnaround period (post-1997) of the ozone evolution  (Ball et al., 2017).

The long-term evolution of the dependent variable excluding the effects of several independent proxies is characterized in
DLM by the "trend term" or background level. Consequently, we extracted the background level from the DLM output. We inferred the posterior distributions on the background level by the Markov chain Monte-Carlo sampling  (Alsing, 2019). The DLM was applied for each individual ensemble member of both experiments, using appropriate proxies for each calculation. We have drawn 200 samples from DLM results, which describe the uncertainty in the posterior distribution. The resulting trends were estimated from the sample mean background levels at each grid point by the Mann-Kendall test for the entire
2015-2099 period, as well as for several sub-periods: 2015-2039, 2040-2069, and 2070-2099, respectively. It was done to properly trace the evolution of trends during the considered period. Then, the trend estimates from all individual ensemble members are averaged to get the mean trends in the ensemble of each experiment. The statistical significance of the calculated ensemble mean trend is estimated by applying the Student's t-test using the standard deviation of trends between individual ensemble members.



# 3 Results

## 3.1 Evolution of drivers of ozone change

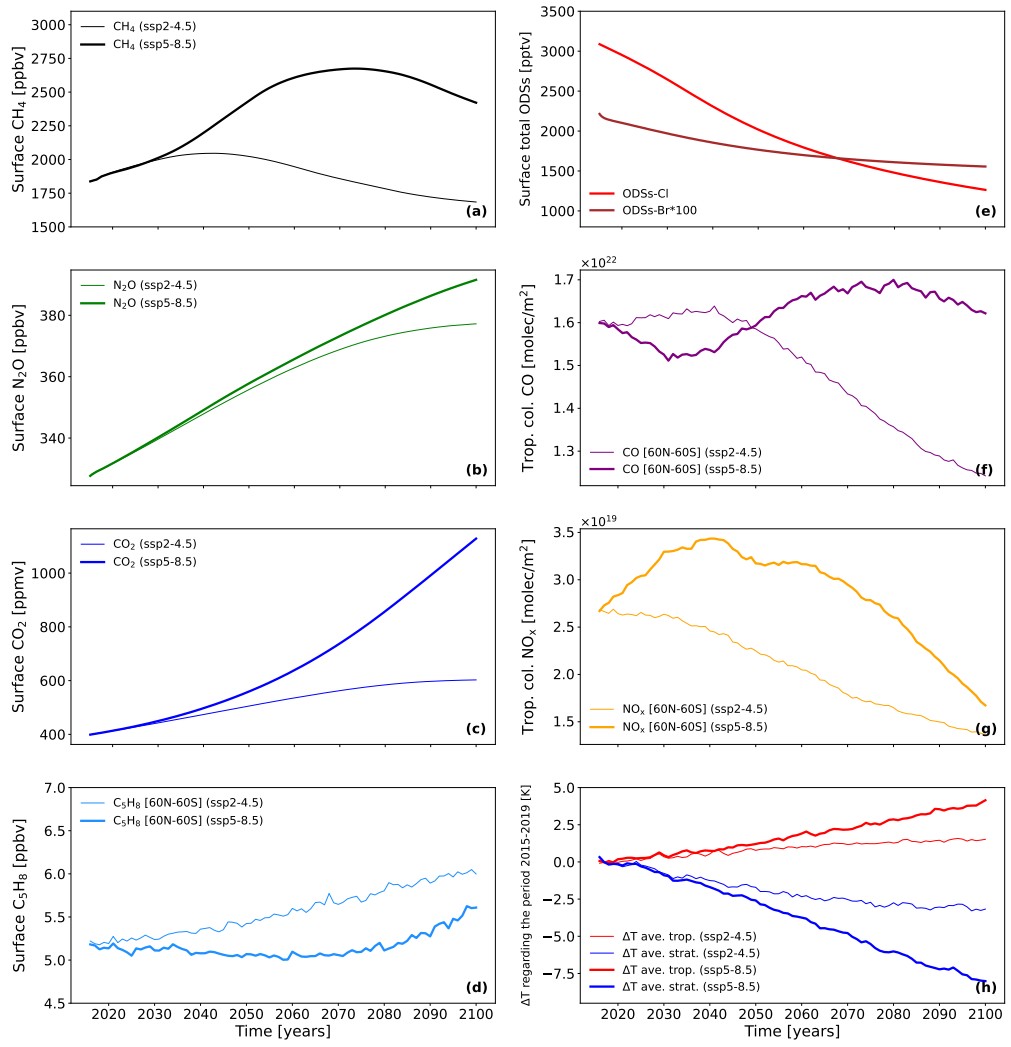

**Figure 1.** Annual mean evolution of drivers of ozone changes between 2015 and 2099 from both SSP2-4.5 (faded lines) and SSP5-8.5 (bold lines) (except for ODSs, since their amounts are identical in both scenarios). This includes **(a)** global surface methane ($CH_4$) concentration [ppbv]; **(b)** global surface nitrous oxide ($N_2O$) concentration [ppbv]; **(c)** global surface carbon dioxide ($CO_2$) concentration [ppmv]; **(d)** near-global [60$^o$N–60$^o$S] surface isoprene ($C_5H_8$) concentration [ppbv]; **(e)** surface total organic chlorine- (red line) and bromine- (x 100) (dark red line) ODSs concentrations [pptv]; **(f)** near-global [60$^o$N–60$^o$S] tropospheric carbon monoxide (CO) column [molecules x cm$^{-2}$]; **(g)** near-global [60$^o$N–60$^o$S] tropospheric nitrogen oxide ($NO_x$) column [molecules x cm$^{-2}$]; **(h)** global mean changes in average tropospheric (red line) and mean stratospheric (blue line) temperature ($\Delta T$) regarding the period 2015-2019



The temporal evolution of several critical drivers of ozone changes is displayed in Figure 1 and demonstrates a considerable difference between SSP2-4.5 and SSP5-8.5 scenarios. $CH_4$ starts to decrease in the mid-2040s in SSP2-4.5, whereas it occurs only in the 2070s under SSP5-8.5. Since CO is partially a product of methane, its evolution in the lower atmosphere resembles the change in $CH_4$, but with a decrease during the first decades in SSP5-8.5. In contrast, near-global $NO_x$ in SSP5-8.5 increases during this period, and after 2040 it starts to decrease. Yet, under SSP2-4.5 $NO_x$ gradually decreased after the 2030s. As such, the decline in tropospheric $NO_x$ and CO columns relates to the air quality change and decline in $CH_4$. In addition, $NO_x$ in the troposphere is produced by lightning activity and airplanes, i.e., future changes in convective activity due to climate change and the growth of aircraft use may contribute to $NO_x$ production. The resilient increase in $CO_2$ and $N_2O$ is observed in both scenarios, with a higher and abrupt increase in SSP5-8.5 but with a sharp slowdown in the growth of $CO_2$ and $N_2O$ concentrations in the last decades of the century, according to SSP2-4.5. chlorine-containing ODSs (red line in **(e)** panel of Figure 1) are decreasing throughout the whole period. In its turn, a decline in bromine-containing hODSs (dark red line in **(e)** panel of Figure 1) is decelerated by the end of the century. Biogenic isoprene ($C_5H_8$) evolves with a steady increase in SSP2-4.5, whilst in SSP5-8.5 $C_5H_8$ decreases till the 2060s and slightly increases by the end of the century. The global mean temperature changes relative to the present-day period show a stable increase in mean tropospheric and a decrease in mean stratospheric temperatures with a more intense change under SSP5-8.5. Under SSP2-4.5, the temperature changes become less pronounced in the late century due to a significant slowdown in GHG growth.





## 3.2 Ozone anomalies for the period 2015-2099 relative to the present-day ozone concentration

**Figure 2.** Near-global [60°N-60°S] annual mean anomaly ($\Delta O_3$) of column ozone and DLM fits (both in Dobson Units, DU) between 2015 and 2099, presented regarding the mean $O_3$ for the 2015-2019 period. Red line: $\Delta O_3$ under SSP2-4.5 scenario; Blue line: $\Delta O_3$ under SSP5-8.5 scenario. $\Delta O_3$ presented for **(a)** mesosphere; **(b)** upper stratosphere; **(c)** middle stratosphere; **(d)** lower stratosphere; **(e)** entire model atmosphere; **(f)** entire stratosphere, and **(g)** troposphere. Shadings represent the 2-$\sigma$ standard deviation between ensemble members of the experiment.





Figure 2 shows the annual mean partial and total column ozone changes in the near-global region throughout the 21$^{st}$ cen-
tury with respect to the period 2015-2019 in different atmospheric layers. We calculated changes in $O_3$ relative to the period
2015-2019 to estimate the future modeled ozone change regarding its current concentration. It was noted that the evolution of
tropospheric ozone is largely determined by changes in $CH_4$, CO, and $NO_x$ (see Figure 1). The contribution of $NO_x$ seems
to play a larger role in SSP2-4.5 due to the less abundant $CH_4$ and CO. Nevertheless, in SSP5-8.5, the sharp decrease in $O_3$,
starting after 2065 resulted mainly from the decrease in $NO_x$, as both $CH_4$ and CO start to decrease later. On the other hand,
the projected sharp decline in $NO_x$ makes CO a more important driver of tropospheric ozone evolution in the last part of the
century, despite it also slightly decreasing. Note that the decline in tropospheric ozone content in SSP2-4.5 starts in the 2030s,
much earlier than in SSP5-8.5. A steady increase in tropospheric ozone in SSP5-8.5 is observed by the 2060s and afterwards
starts to sharply decrease during the last decades of the century, similarly to the RCP6.0 scenario (Revell et al., 2015b). Albeit,
in the late century, tropospheric ozone will be lower than it is now in both scenarios. Yet, the difference in the zero-crossing
point time is about 45 years between scenarios. The tropospheric ozone content becomes lower than present-day in SSP2-4.5
already in 2045, while in SSP5-8.5, it is lower only after 2090.

The lower stratospheric ozone on a near-global scale shows signs of increasing until mid-century. However, over the last half
of the century, it began to gradually decline, showing a reduction of about -1.5 DU in SSP2-4.5 and -3.5 DU in SSP5-8.5 by
2099. In fact, this ozone decrease is mainly induced by the intensification of transport from the tropics toward the mid-latitudes.
In addition, the decline in averaged ozone over 60°N-60°S indicates that the tropical ozone decrease in the lower stratosphere
starts to prevail over the ozone recovery from the effects of hODSs on a near-global scale, as seen in Figure 2. In contrast, $NO_x$
might still contribute to ozone production in the lower stratosphere. And when it starts to decline, the ozone abundance also
decreases stronger. In the middle and upper stratosphere, ozone recovers throughout the period due to a decline in hODSs level,
with a growth of 1 DU (in the middle stratosphere) and 3.5 DU (in the upper stratosphere) according to SSP2-4.5 and about
3 DU (in the middle stratosphere) and 6.5 DU (in the upper stratosphere) according to SSP5-8.5 by the end of the century. In
SSP5-8.5, a much more intense growth after the 2040s is observed, when the discrepancy in $CO_2$ evolution between scenarios
becomes larger (see Figure 1), i.e., the stratospheric temperature is lower in SSP5-8.5. In both scenarios, the evolution of the
near-global averaged mesospheric ozone also increases. By the end of the century, the ozone content in the mesosphere will be
higher by ~0.13 DU under SSP2-4.5 and by 0.27 DU under SSP5-8.5 than its modern level. This larger ozone enhancement in
SSP5-8.5 might be due to lower temperatures and some influence of decreasing $NO_x$ in the mesosphere. Even so, extra-polar
mean total column ozone content by the end of the century will be definitely higher than presently, wherein the magnitude of
the increase is three times higher in SSP5-8.5 than in SSP2-4.5.




## 3.3 Ozone and drivers trends development during the period 2015-2099

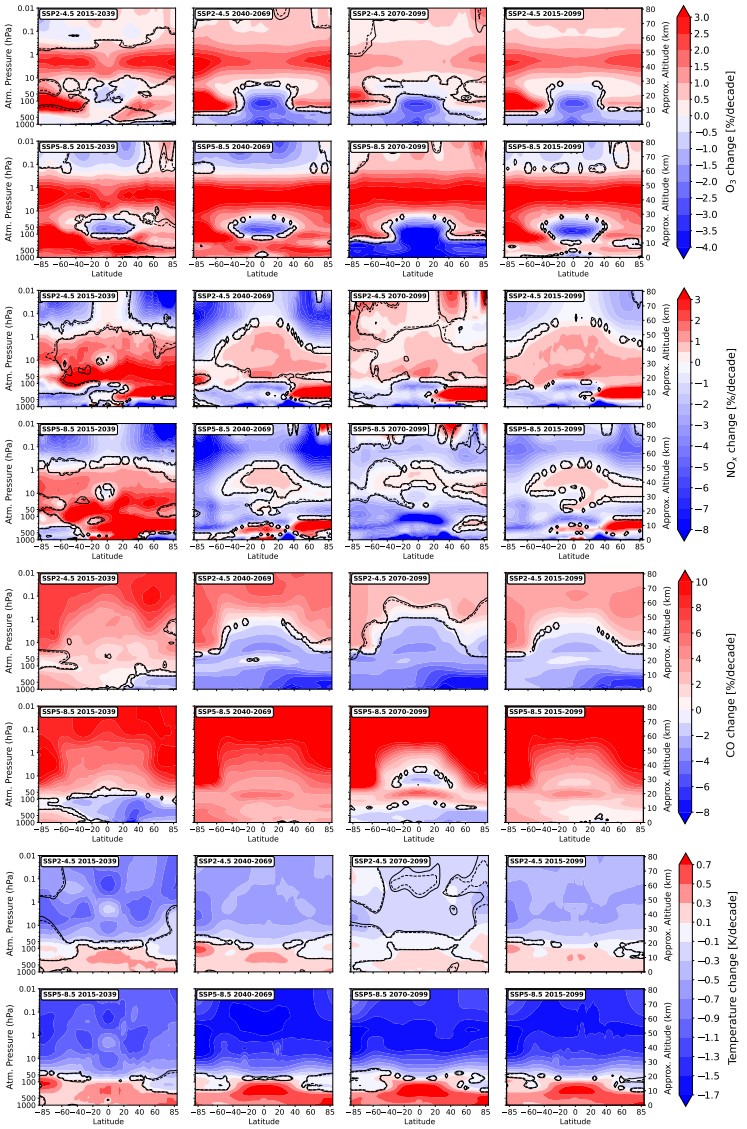

**Figure 3.** Profiles of trends in $O_3$, $NO_x$, CO, and temperature for the 2015-2099 period, and different sub-periods from both SSP2-4.5 and SSP5-8.5 simulations. The name of the corresponding scenario and the period are indicated in the upper left corner of each panel. The dashed line is the delimiter of the region with significance at the 90% level for positive or negative changes; the solid line is the same at the 95% level.





The understanding of ozone evolution requires knowing the changes in the driving agents such as temperature and important gas species involved in the ozone production/destruction cycles. The evolution of the CO, $NO_x$, temperature, and $O_3$ trends between 2015 and 2099 from both experiments is presented in Figure 3.

Carbon monoxide is produced via $CO_2$ photolysis by solar irradiance in the upper atmosphere (e.g., Karagodin-Doyennel
et al., 2021) and can be transported down mostly over the high latitudes during cold seasons. Therefore, its abundance in these areas strongly reflects $CO_2$ behavior mimicking a steady increase in SSP5-8.5 and stabilization in SSP2-4.5. The increase in CO in the stratosphere should not strongly contribute to the ozone changes, however, some slight effect can be expected from the removal of OH caused by $CO + OH \rightarrow CO_2 + H$ reaction (Wofsy et al., 1972). In the troposphere, the CO source is driven by methane and biogenic VOCs. Therefore, we observe a steady CO decline after 2040 in SSP2-4.5 following the drop in
methane emissions (see Figure 1). For the SSP5-8.5 the change of signs appears in 2070 after flattening and a small decline of the methane mixing ratio. An initial negative tendency for the 2015-2039 period is related to a small decrease in VOCs. The CO tendencies in the stratosphere are defined by the upward transport and mixing of the tropospheric air. Carbon monoxide can be considered as a proxy for the level of organic species, which are a necessary part of the tropospheric ozone production mechanism. The concentration of $NO_x$ is the second part participating in this process.


In the mesosphere, $NO_x$ ($NO + NO_2$) is mostly produced by $N_2O$ oxidation, energetic particles, and influx from the thermosphere. They can be destroyed by solar irradiance via NO photolysis followed by cannibalistic $N + NO \rightarrow N_2 + O$ reaction. Because the thermospheric source is the same for both cases and is partly accounted for by solar proxies, the $NO_x$ trend in the mesosphere depends on the available $N_2O$ and temperature, which regulate the efficiency of the cannibalistic reaction, making
it faster for the cooler environment in the future. Despite a steady $N_2O$ increase (see Figure 1) the $N_2O$ in the mesosphere is less available due to its higher destruction by enhanced ozone and $O(^1D)$ concentration in the stratosphere. Thus, less $N_2O$ abundance and cooler temperature lead to a general decrease of the mesospheric $NO_x$. For the 2070-2099 period, however, the $NO_x$ depletion for the SSP2-4.5 case is not so pronounced due to probably very small mesospheric cooling.

Stratospheric $NO_x$ concentration is mostly regulated by the production via $N_2O + O(^1D) \rightarrow NO + NO$ and conversion to reservoir species which depends on the temperature and availability of hydrogen and halogen-containing species, which deactivate $NO_x$ building reservoir species like $HNO_3$ or $ClONO_2$. Therefore the stratospheric $NO_x$ increase is more substantial in the SSP2-4.5 case when the cooling and water vapor increases are not so pronounced as in the SSP5-8.5 case.

$NO_x$ is mostly declining in the lower troposphere due to improved air quality. Also, most periods in both scenarios show the permanent increase in free tropospheric $NO_x$ over the Northern Hemisphere upper troposphere that is maintained by aircraft emissions.





The temperature trend patterns are rather expected. Continuous increase of greenhouse gases leads to tropospheric warming
and stratospheric cooling (e.g., Lee et al., 2021) and both are substantially more pronounced in the SSP5-8.5 scenario due to
more intensive anthropogenic activity. The tropospheric warming this case is more prominent over the Northern Hemisphere
due to Arctic amplification (e.g., Previdi et al., 2021) and in the lower stratosphere over the southern high latitude where the
ozone concentration is increasing due to the recovery from the halogen loading to pre-ozone hole conditions. The radiative
cooling by greenhouse gases dominates in the stratosphere over some warming caused by the stratospheric ozone increase
and agrees with their time evolution shown in Figure 1. During the first 2015-2039 sub-period, the quadrupole structure of
stratospheric temperature trends is observed in both scenarios, which is dynamically induced (Ball et al., 2016), and is barely
observed in later sub-periods.

The ozone change patterns substantially differ between layers. In the troposphere, the ozone decrease is observed for the
SSP2-4.5 scenario starting from 2040 as well as for the entire period. This behavior is explained by the continuous decrease of
the ozone precursors related to the improvement of air quality. For the SSP5-8.5 similar process occurs only after 2070 when
$NO_x$ atmospheric abundance decline is the most prominent (see Figure 1(**g**)). A similar decrease in the tropospheric ozone
resembles the results obtained by Revell et al. (2015b) using the RCP6.0 scenario. Some increase in $NO_x$ level before 2070
leads to positive tropospheric ozone trends, which makes ozone trend positive for the entire period.


In the upper stratosphere and southern lower stratosphere, the ozone increase is very persistent because it is driven by a
steady decline of the halogen loading (see Figure 1). The ozone increase in the upper stratosphere is stronger for the SSP5-8.5
case because more pronounced stratospheric cooling leads to less intensive catalytic ozone destruction cycles. Another area
with a persistent trend appears in the tropical lower stratosphere, where intensified in the warmer climate Brewer-Dobson cir-
culation drives negative ozone trends (e.g., Zubov et al., 2013). This feature is more pronounced for the SSP5-8.5 scenario
after 2070 because of the stronger warming. Before 2070 and for the entire period the magnitude of the ozone decline in this
area is virtually the same for both cases due to compensation of the dynamical loss by increased tropospheric ozone obtained
for SSP5-8.5.

In the upper mesosphere, ozone decreases until 2070 under SSP5-8.5 due to an increase in $CH_4$ causing an increase in meso-
spheric abundance of $H_2O$ and, hence, an enhancement of $HO_x$ radicals. Under SSP2-4.5, mesospheric ozone has generally
increased over the entire period of 2015–2099, since $CH_4$ only slightly increases until the 2040s and then begins to decline.

### 3.4   Total column ozone trends development during the period 2015-2099

In one way or another, changes in ozone in different layers of the atmosphere contribute to a change in total column ozone. It
is essential for humanity to know the future evolution of total ozone because it affects changes in ground-level UV radiation.
Figure 4 shows the evolution of trends in total column ozone as a function of month and latitude over the period 2015-2099



and sub-periods.

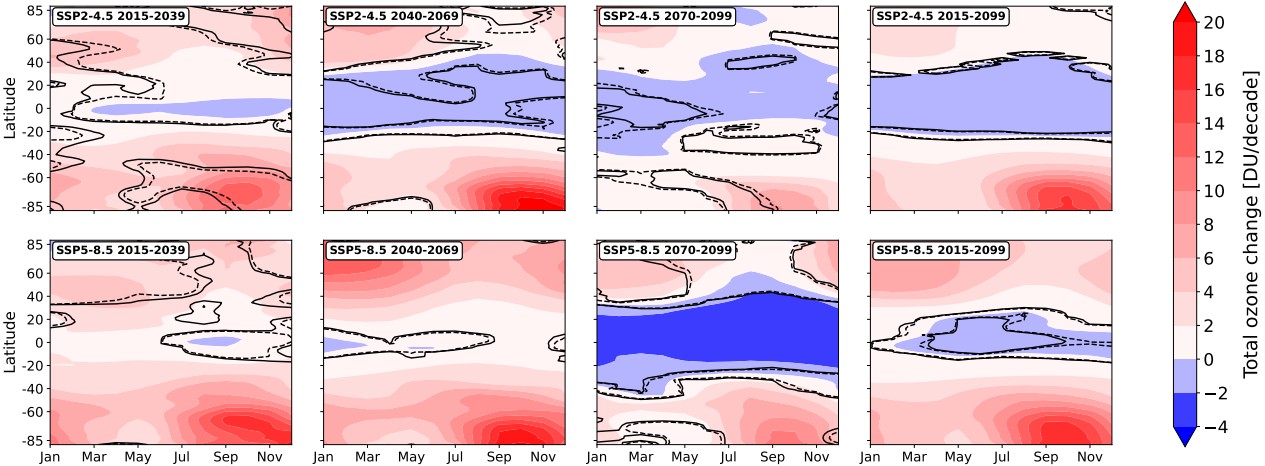

**Figure 4.** Trends in total column ozone as a function of month and latitude for the 2015-2099 period, and different sub-periods from both SSP2-4.5 and SSP5-8.5 simulations. The name of the scenario and the period for which the trends are calculated are indicated in the upper left corner of each panel. The dashed line is the delimiter of the region with significance at the 90% level for positive or negative changes; the solid line is the same at the 95% level.

The total ozone recovery in austral spring over the Southern Hemisphere is generally similar between both scenarios since it
is driven by the phase-out of hODSs emissions, which are identical in both scenarios. However, the ozone increase is slightly higher in SSP5-8.5, owing to a lower temperature in the stratosphere. It is also seen that during 2070-2099, in both scenarios, the ozone increase is slowed down. This might be because of the slower hODS decline (see Figure 1). In mid-latitudes, the ozone increase is also higher in SSP5-8.5 due to both temperature and more intense transport from the tropics. In contrast, in the tropics, trends in total ozone largely differ between scenarios. In SSP2-4.5, the tropical total ozone tends to reduce during
the entire period by about -2 DU/decade due to ozone decrease in the lower stratosphere and troposphere. A strong decline in total ozone of about -4 DU/decade between 2070 and 2099 is observed in SSP5-8.5 but in other sub-periods, the trend in tropical total column ozone is generally near-zero due to an increase in tropospheric ozone that partly compensates for the ozone decline in the lower stratospheric ozone. During boreal spring, the total ozone also increases in the Northern Hemisphere in both scenarios, with a higher increase in SSP5-8.5.


Thus, some increase in surface UV level over the tropics and a decrease over the middle and high latitudes can be expected in both scenarios, but in SSP2-4.5 it will be higher in the tropics throughout the entire period, and in SSP5-8.5 only in the late century. On the contrary, the decrease in surface UV level at middle and high latitudes is expected to be greater in SSP5-8.5 than in SSP2-4.5 due to higher ozone.





## 4 Discussion and conclusions


In this paper, we have evaluated atmospheric ozone trends based on two sets of ensemble simulations using SOCOLv4 covering the period from 2015 to 2099. One simulation is based on the SSP2-4.5 scenario and the other is based on the SSP5-8.5 scenario, which differs in greenhouse gas emissions and ozone precursors. The trends in ozone as well as in non-hODSs drivers of ozone evolution such as $NO_x$, CO, and temperature are derived using DLM. The ozone layer is expected to increase on a

near-global scale throughout the entire century, because of the ban on the production of hODSs by the Montreal Protocol. However, the evolution of atmospheric ozone in different atmospheric layers differs greatly between the two SSP scenarios. The tropospheric ozone evolution, driven mainly by CO and $NO_x$ changes, shows a difference in the time of inflection point when tropospheric ozone begins to decline. In SSP2-4.5, it began to be observed after 2040, while in SSP5-8.5 it is after 2060. In the mesosphere, upper and middle stratosphere, a resilient increase in ozone is about two to three times higher in SSP5-8.5

than in SSP2-4.5 since the temperature of these regions in SSP5-8.5 since the negative temperature trends in these regions in SSP5-8.5 are more than $1^{\circ}$ K/decade stronger that retards the catalytic ozone loss. In the lower stratosphere, the near-global ozone content tends to decline after 2040 in SSP2-4.5 and after 2050 in SSP5-8.5, but by the end of the century, the decrease in SSP5-8.5 is about three times higher due to faster meridional transport of ozone to the poles.

In general, it is difficult to establish trends in ozone in the extratropical lower stratosphere due to the large uncertainty associated with natural variability (Ball et al., 2018). However, for the long-term periods, statistically robust projection in this part of the atmosphere is possible, as we show in this study. In addition, we show that some recently discovered factors, like unregulated ODSs and oceanic VSLSs, which were suspected to affect the ozone layer evolution did not lead to serious changes in expected ozone layer evolution for the used GHGs scenarios. Note that it might be also because their negative influences are

compensated to some extent by positive effects from greenhouse gases. Also, we have no future volcanic activity considered in our study because it is hard to predict volcanic eruptions. However, severe implications for the ozone layer in the future are expected if strong volcanic eruptions occur (Klobas et al., 2017). In addition, no less important for the future ozone evolution might be the projected decline in solar activity throughout the 21$^{\text{st}}$ century (Steinhilber and Beer, 2013; Matthes et al., 2017). It is well known that solar activity mainly drives photochemical and dynamical processes in the stratosphere and is responsible

for the ozone formation and radiation budget (Haigh, 1994; Rozanov et al., 2004; Hood and Soukharev, 2003; Egorova et al., 2004). Therefore, a decline in solar activity might lead to a decrease in atmospheric ozone production, causing some negative implications for its future evolution (Anet et al., 2013; Rozanov et al., 2016; Arsenovic et al., 2018).

Nevertheless, total column ozone is expected to increase almost everywhere, except in the tropics. In both polar regions,

the total ozone increases with a slightly higher intensity in SSP5-8.5. In the mid-latitudes, the total ozone is also increasing thanks to upper stratospheric ozone increase and transport from the tropics. Conversely, in the tropics, it generally declines in SSP2-4.5 due to both tropospheric and lower stratospheric ozone decrease showing about -2 DU/decade; it barely changes in SSP5-8.5 with a sharp decrease of about -4 DU/decade only during the last decades of the century due to a severe reduction in

both tropospheric and lower stratospheric ozone content. We showed that besides changes in the stratospheric ozone column, the tropospheric column ozone evolution is also essential to be considered since it may seriously contribute to total column ozone evolution, especially in the tropics.

A much stronger ozone increase in the upper part of the middle atmosphere and mid-to-high latitudes of the lower stratosphere might also be expected in the SSP5-8.5 scenario. In this regard, it may seem that the "more greenhouse gases" scenario is better because, despite higher near-surface temperatures, it will be more favorable for ozone increase over the most populated areas. However, the excessive increase in ozone over mid-to-high latitudes may also cause negative consequences for human well-being. Exceeding the required level of total ozone content, especially over the most inhabited areas, means more UV absorption and consequently less surface level of UV radiation, than required for human health. It causes less vitamin D synthesis and therefore increases the risk of diseases related to vitamin D deficiency, like rickets and osteomalacia (Butler et al., 2016). In addition, it is worth paying attention to the evolution of ozone in the tropics. There is a risk of a decrease in total ozone content leading to an increase in surface UV level abnormally, which also causes negative effects on human health, like an increased risk of skin cancer and cataracts (Butler et al., 2016).

The important message in this regard is to find a way to bring the ozone content in the atmosphere to an equilibrium state when it is neither lower nor higher than necessary. Thus, we emphasize that the findings presented in this study will be useful for further improvement of socioeconomic pathways policies to determine the route to maintain the global total ozone content favorable for the sustainable development of human civilization.

*Acknowledgements.* A.K.-D., E.R., T.S., T.E., and J.S. are grateful to the Swiss National Science Foundation for supporting this research through the №200020-182239 project POLE (Polar Ozone Layer Evolution). The work of E.R. and T.S. has been partly performed in the SPbSU "Ozone Layer and Upper Atmosphere Research" laboratory, supported by the Ministry of Science and Higher Education of the Russian Federation under agreement 075-15-2021-583. Calculations were supported by a grant from the Swiss National Supercomputing Centre (CSCS) under projects S-901 (ID 154), S-1029 (ID 249), and S-903.

*Data availability.* The SOCOLv4 simulations can be accessed upon request from https://doi.org/10.5281/zenodo.7318315 (Karagodin-Doyennel , 2022)

.





*Author contributions.* AKD processed the data, visualized the results, and prepared the original draft. ER and TP supervised this research. ER originated the idea for this study. TS, JS, and AKD designed the experiments and performed simulations. All authors participated in editing the manuscript and discussing the results.

*Competing interests.* The authors declare that they have no conflict of interest.



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





## Appendix A

**Figure A1.** Input quantities (proxy variables) for the forcing of the SOCOLv4 simulations. Fade colors: SSP2-4.5; Bright colors: SSP 5-8.5.





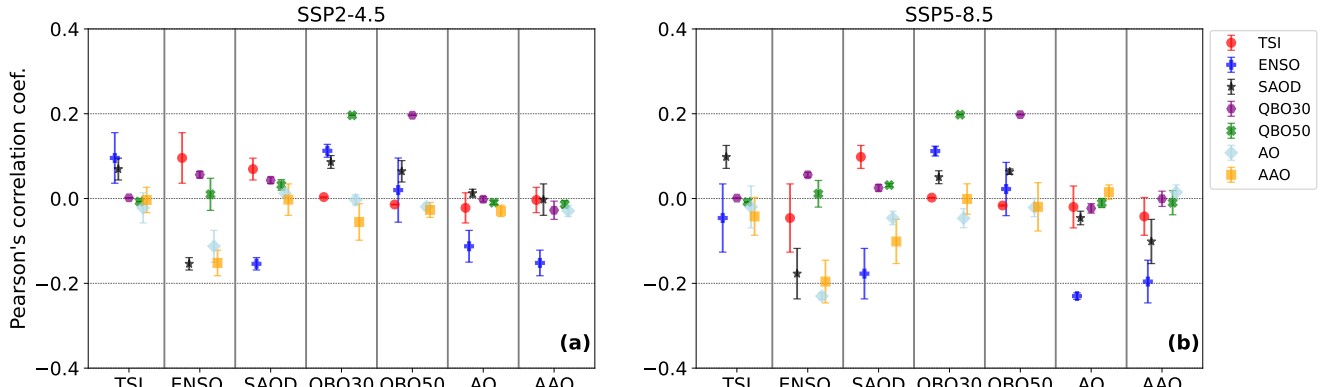

**Figure A2.** Pearson's linear correlation coefficients for different covariate variables (proxy variables) of the SOCOLv4 simulations for the 2015-2099 period: **(a)** for SSP2-4.5; **(b)** for SSP5-8.5. Error bars represent the 2-σ standard deviation of correlation coefficients between ensemble members.