# Peer review of "The future ozone trends in changing climate simulated with SOCOLv4"

_EGUsphere, 2022_

## Author Comment (AC1)

*Dear referee,*

*We appreciate you for reading and reviewing our paper and for your valuable suggestions and comments to improve the manuscript in a good way. The detailed response to each comment is provided below (in red).*

Review for Karagodin-Doyennel et al., "The future ozone trends in changing climate simulated with SOCOLv4", submitted to ACP

Here authors use the SOCOLv4 Earth System Model (ESM) to simulate future ozone evolution. Results from two model simulations (based on two potential Shared Socioeconomic Pathways (SSP) scenarios: SSP2-4.5 and SSP5-8.5) are presented. SOCOL_v4 predicts a decline in tropospheric ozone around the 2030s for SSP2-4.5 and after 2060s under SSP5-8.5 scenarios primarily due to decrease in ozone precursors such as NOx and CO. As expected, models also predict ozone increase in the upper/ middle stratosphere globally and high-latitude lower stratosphere. Model also predicts that under SSP5-8.5, the stratospheric ozone increases are largely due to stronger stratospheric cooling as more greenhouse gases lead to more cooling that slows ozone loss. On the other hand, both the simulations predict ozone decrease in the tropical lower stratosphere as strengthening of BD circulation transports more ozone to the mid-high latitudes.

As confirmed by various recent studies, the model does not predict any decrease in lower stratospheric ozone at mid-latitudes. Consistent with our understanding, SOCOLv4 predicts an increase in stratospheric ozone in the 21st century due to decreases in the ODS and increases in the GHGs.

Overall, this is a well organised manuscript with some room for improvements. So I would recommend the manuscript for the publications if authors can address minor comments suggested below

1. I am struggling to find clear motivation. What are the largest sources of uncertainties in our understanding about the future evolution of ozone layer . What was done in earlier SOCOL studies and what was missing and how this manuscript is able to improve those biases. There are serveral papers like Morganstern et al., Dhomse et al., Keeble et al, highlighting the role of GHG in explaining decreases in tropical ozone. So, what is new here?

Thank you for pointing this out. We agree that the motivation for this study has not been formulated as clearly as it should be in the introduction. Yes, the ozone evolution in the future has been continuously investigating using CCMs, starting from the Eyring et al., (2007) study. Yet, many other studies have been carried out since then because CCMs as well as the future GHG scenarios have been continuously developing and upgrading. Thus, in the light of ongoing model development, each new study gives additional insight into the future ozone evolution and using up-to-date statistical methods allows drawing a more accurate conclusion on the future ozone trends.

Here, we analyze the ozone simulations performed for the 2015-2099 period with the new Earth System Model SOCOLv4, based on two SSP (2-4.5 and 5-8.5) scenarios. The SOCOLv4 has a lot of ozone-related upgrades against its previous version, SOCOLv3

(that was previously used in similar studies), as it was mentioned in the Sukhodolov et al., (2021). Basically, it is the new generation of models that includes several essential components of the Earth system, which now interactively coupled to each other (Sukhodolov et al., 2021).

In our study, the new-generation statistical approach, namely, Dynamical Linear Modelling (DLM) used to retrieve and evaluate the future ozone evolution from the SOCOLv4 simulations. The DLM has not been utilized before in regression analysis of the future ozone evolution; in previous studies, the conventional multi-linear regression was used. The DLM has several advantages over the MLR (see in the response to the comment 4 for details) and it is preferable to use DLM due to non-homogeneity of ozone changes in different atmospheric regions.

Also, the problem remains open regarding the lower stratospheric ozone evolution in the mid-latitudes that is questioned due to signs of ozone decline in this region found in ozone observational composite using DLM (Ball et al., 2018). So, it is also important to continue studying this region but for the future using a similar statistical tool, that was done in the study about the past ozone trends in SOCOLv4 (i.e., Karagodin-Doyennel et al., 2022).

In addition, the important advantage of our study is that we show the ozone evolution, its chemical forcings (like $NO_x$ and CO), temperature vertical profile changes as well as latitude-month total column ozone trend for different sub-periods within 21$^{st}$ century. Previously, the transient trends have not been demonstrated in a such way. In the multi-model study of Morgenstern et al. (2018), the analysis of GHGs from RCP forcing of ozone was done for the period 1960–2100, without profile trend evolution during the 21$^{st}$ century. Dhomse et al., (2018) performed the analysis of various CCMI simulations under several CMIP5 RCP forcings to estimate the return dates for future total and partial ozone column evolutions from anthropogenic halogens. The analyzed ozone evolutions were averaged for specific geographical regions. Despite the detailed analysis of future ozone evolution, the robust statistical test of these results was not performed. The resent multi-model study of Keeble et al., (2021) analyzed the long-term total and partial ozone column evolution over the 1960-2100 period using CMIP6 simulations under SSP emission scenarios. In particular, this study presents estimates of ozone profile change and total column ozone between the end of 21$^{st}$ century and present time, similarly as it was done in our study. However, the results provided without performing a robust statistical analysis that would show in which regions trends can be considered as robust but not artificially induced and where the statistical significance of estimated trends is low. This proposed to be done using regression analysis by excluding the effect of natural variability that allow performing an accurate statistical significance evaluation of trends induced by GHG and ODSs. In addition, the evolution of ozone trend profiles and latitude-month total column ozone during 21$^{st}$ century also has not been addressed in this study.

We should state the importance of showing the trends separately for subperiods as this allowed extracting more detailed information on how modelled ozone trend and dependencies evolve during the early, middle, and late 21$^{st}$ century under different GHG forcings. It is essential especially in the contest of clarity of ozone prediction for the sub-century scale.

In addition, SOCOL model has not been included in the previous studies about future ozone trends under SSP scenarios (Keeble et al., 2021 and Sheng et al., 2021) as there were no CMIP6 simulations from SOCOLv4 available that time.

The discussion about the mesospheric ozone evolution was also not complete in previous studies based on SSP scenarios. In our study, we also included the analysis of the mesospheric ozone evolution during the 21$^{st}$ century under SSP scenarios. We show important findings for the mesospheric ozone evolution, related to the change of GHGs concentrations. In addition, the trends in chemical drivers of ozone evolution (as of CO and NO$_x$) were not addressed, besides the atmospheric H$_2$O future change that was described well in Keeble et al., (2021).

Nevertheless, our results on ozone trends agree with multi-model mean from Keeble et al., (2021). We added the comparison to the paper. In addition, a comparison with future ozone trends from several CCMs presented in Sheng et al., (2021) was also done for the troposphere and stratosphere.

Reference:

Eyring, V., Waugh, D. W., Bodeker, G. E., Cordero, E., Akiyoshi, H., Austin, J., Beagley, S. R., Boville, B., Braesicke, P., Bru˙hl, C., Butchart, N., Chipperfield, M. P., Dameris, M., Deckert, R., Deushi, M., Frith, S. M., Garcia, R. R., Gettelman, A., Giorgetta, M., Kinnison, D. E., Mancini, E., Manzini, E., Marsh, D. R., Matthes, S., Nagashima T., Newman, P. A., Nielsen, J. E., Paw- son, S., Pitari, G., Plummer, D. A., Rozanov, E., Schraner, M., Scinocca, J. F., Semeniuk K., Shepherd, T. G., Shibata, K., Steil, B., Stolarski, R., Tian, W., and Yoshiki, M.: Multimodel projections of stratospheric ozone in the 21st century, J. Geophys. Res., 112, D16303, doi:10.1029/2006JD008332, 2007.

Sukhodolov, T., Egorova, T., Stenke, A., Ball, W. T., Brodowsky, C., Chiodo, G., Feinberg, A., Friedel, M., Karagodin-Doyennel, A., Peter, T., Sedlacek, J., Vattioni, S., and Rozanov, E.: Atmosphere-ocean-aerosol-chemistry-climate model SOCOLv4.0: description and evaluation, Geosci. Model Dev., 14, 5525–5560, https://doi.org/10.5194/gmd-14-5525-2021, 2021.

Keeble, J., Hassler, B., Banerjee, A., Checa-Garcia, R., Chiodo, G., Davis, S., Eyring, V., Griffiths, P. T., Morgenstern, O., Nowack, P., Zeng, G., Zhang, J., Bodeker, G., Burrows, S., Cameron-Smith, P., Cugnet, D., Danek, C., Deushi, M., Horowitz, L. W., Kubin, A., Li, L., Lohmann, G., Michou, M., Mills, M. J., Nabat, P., Olivié, D., Park, S., Seland, Ø., Stoll, J., Wieners, K.-H., and Wu, T.: Evaluating stratospheric ozone and water vapour changes in CMIP6 models from 1850 to 2100, Atmos. Chem. Phys., 21, 5015–5061, https://doi.org/10.5194/acp-21-5015-2021, 2021.

Shang, L., Luo, J., and Wang, C.: Ozone Variation Trends under Different CMIP6 Scenarios, Atmosphere, 12, 112, https://doi.org/10.3390/atmos12010112, 2021.

2. Why there is a paragraph about the VSLS if there are no plots showing its impact on the ozone layer?

The updated boundary conditions for VSLSs were used for the SOCOLv4 simulations, performed for this study. This is one of the factors that might affects the accuracy of future ozone evolution. Nevertheless, in this study, we had no plans to differentiate the effect of VSLS on the ozone evolution from others.

We agree that the huge paragraph in the introduction about new VSLSs can be omitted. Therefore, we decided to exclude this paragraph from the introduction, but keep some information about the updated VSLSs in the model description section.

Can you explain why sAOD term is included in DLM model though there is no volcanic eruption in the simulations

Here, used DLM set-up is the same as it was used to analyze the past ozone trends in SOCOLv4 (see Karagodin-Doyennel et al., 2022). Indeed, compared to the past period, the future simulations do not account for any future volcanic eruptions. However, since in SOCOLv4, the aerosol fields are not prescribed but interactively calculated, the future SAOD is also changing with a time and have a difference between considered SSP scenarios. This difference caused by temperature and atmospheric dynamics variations, induced by GHGs. In the Figure A1 provided in the Appendix, the simulated SAOD changes with a slight trend over the 21$^{st}$ century and there is a difference in SAOD between considered SSP scenarios. It should be noted that the robust statistical methods, like DLM, might be sensitive to even small changes, i.e., the resulting statistical significance might be violated if this variable (like SAOD) is not included to the analysis. Thus, it is important to consider the SAOD as a regressor to analyze simulations from the model where the advanced aerosol microphysics is included and aerosol fields are not prescribed.

3. Line 198: Usage of DLM to model simulated data is still unclear. Please include some clear information explaining why this type of analysis should use DLM rather than multivariate (or ordinary least square) regression or simpler composite analysis. I strongly feel that using DLM for the analysis of observational data is OK as we have just one realisation about the past atmosphere. But as you have 3 ensemble members for each type of simulation, does DLM provide unique insight in model world compared to simple averaging and smoothing?

Yes, the DLM method has important advantages over the conventional multi-linear regression (MLR) and allow obtaining more robust trend estimates, especially if time series have non-homogenized time-varying trends in different atmospheric regions. Thus, using the DLM to obtain ozone trend estimates is desirable. The main advantages are listed in several previous studies, i.e., Laine et al. (2014), Ball et al., (2018), Alsing (2019) etc.. Generally speaking, DLM is much more flexible method than MLR largely because of the following advantages: the background trend is allowed to vary in time; regression coefficients are not fixed but may slowly vary in time that allow capturing more variability from the time-series; auto-regressive process is inferred together with other DLM parameters that decrease its uncertainty; account for the non-constant error distribution in the regression coefficients estimation. Laine et al. (2014) mentioned that if trends are non-linear, the estimates from DLM are expected to be more robust and found that the trends estimated in MLR might be even inverse to those estimated using DLM. In the Bognar et al., (2022) study, the detailed comparisons of MLR and DLM is given, showing that MLR has noticeable limitations against DLM. The DLM is broadly used in the recent ozone studies to analyses the observations and model data but has not been used to evaluate the future ozone evolution.

Reference:

Laine, M., Latva-Pukkila, N., and Kyrölä, E.: Analysing time-varying trends in stratospheric ozone time series using the state space approach, Atmospheric Chemistry & Physics, 14, 9707–9725, https://doi.org/10.5194/acp-14-9707-2014, 2014.

Ball, W. T., Alsing, J., Mortlock, D. J., Staehelin, J., Haigh, J. D., Peter, T., Tummon, F., Stübi, R., Stenke, A., Anderson, J., Bourassa, A., Davis, S. M., Degenstein, D., Frith, S., Froidevaux, L., Roth, C., Sofieva, V., Wang, R., Wild, J., Yu, P., Ziemke, J. R., and Rozanov, E. V.: Evidence for a continuous decline in lower stratospheric ozone offsetting ozone layer recovery, Atmos. Chem. Phys., 18, 1379–1394, https://doi.org/10.5194/acp-18-1379-2018, 2018.

Alsing, (2019). dlmmc: Dynamical linear model regression for atmospheric time-series analysis. Journal of Open Source Software, 4(37), 1157. 2 https://doi.org/10.21105/joss.01157

Bognar, K., Tegtmeier, S., Bourassa, A., Roth, C., Warnock, T., Zawada, D., and Degenstein, D.: Stratospheric ozone trends for 1984–2021 in the SAGE II–OSIRIS–SAGE III/ISS composite dataset, Atmos. Chem. Phys., 22, 9553–9569, https://doi.org/10.5194/acp-22-9553-2022, 2022.

Technical

Line 115: Hu et al. (2015),

It was corrected.

Line 163: Only GHGs (prescribed ODSs are identical)?

Yes, prescribed ODS fluxes are identical between SSP2-4.5 and SSP5-8.5 and the only difference in boundary conditions is GHGs surface level.

---

## Author Comment (AC2)

*Dear referee,*

*We grateful you for reading and reviewing our paper as well as for your valuable suggestions and comments to improve our manuscript. Please find the detailed response to each comment below (in red)*

The MS: "The future ozone trends in changing climate simulated with SOCOLv4" deals with ozone evolution in the "middle-of-the-road" (SSP2-4.5) and "fossil-fueled" (SSP5-8.5) scenarios in the troposphere and middle atmosphere. As the authors have already presented the results from their historical simulations and comparisons to measurements in another paper, this study concentrates only on the future changes.

As expected, ozone is in increasing in the stratosphere and decreasing in the lower stratosphere and troposphere. However, the mesospheric part is interesting as there are not many analyses for this region. This is a well-written MS and I have only some minor comments on this.

Major:

I thought a comparison between the previous RCP scenarios with the latest SSP scenarios is needed in the discussion. There are some studies based on CIMIP 5 ozone results. This should be in the modelling point of view. You have mentioned some in Introduction, but a discussion of the ozone results from both CMIP5 and CIMP6 are needed.

Thank you for this comment. We agree that the comparison with results of future ozone trend analysis in CMIP6 simulations based on SSP2-4.5 and SSP5-8.5, which available in previous studies (i.e., Keeble et al., 2021, Shang et al., 2021) was not complete and should be extend in the discussion of our results. Therefore, we extended the discussion section with this comparison with results from these studies.

However, we should state that adding the direct comparison to previous studies based on CMIP5 RCP or other SSP scenarios in the discussion is not possible, because even in the corresponding scenarios, there is a discrepancy in climate forcings between RCP and SSP scenarios due to difference in GHG concentration pathways, which might cause quite different atmospheric and the ozone response (see Revell et al., 2022). Thus, the direct comparison requires performing additional model experiments with these RCP or others SSP scenarios, but this requires a large computer power budget extension since SOCOLv4 is high computing power-consuming model, but this is out of the scope of this study.

References:

Keeble, J., Hassler, B., Banerjee, A., Checa-Garcia, R., Chiodo, G., Davis, S., Eyring, V., Griffiths, P. T., Morgenstern, O., Nowack, P., Zeng, G., Zhang, J., Bodeker, G., Burrows, S., Cameron-Smith, P., Cugnet, D., Danek, C., Deushi, M., Horowitz, L. W., Kubin, A., Li, L., Lohmann, G., Michou, M., Mills, M. J., Nabat, P., Olivié, D., Park, S., Seland, Ø., Stoll, J., Wieners, K.-H., and Wu, T.: Evaluating stratospheric ozone and water vapour changes in CMIP6 models from 1850 to 2100, Atmos. Chem. Phys., 21, 5015–5061, https://doi.org/10.5194/acp-21-5015-2021, 2021.

Shang, L., Luo, J., and Wang, C.: Ozone Variation Trends under Different CMIP6 Scenarios, Atmosphere, 12, 112, https://doi.org/10.3390/atmos12010112, 2021.

Revell, L. E., Robertson, F., Douglas, H., Morgenstern, O., and Frame, D.: Influence of Ozone Forcing on 21st Century Southern Hemisphere Surface Westerlies in CMIP6 Models, Geophys. Res. Lett., 49, e98252, https://doi.org/10.1029/2022GL098252, 2022.

Minor:

L6: "and upper and middle"

Done

L9: speed up of BDC?

We rewrote it as follows:

Speed-up of the Brewer-Dobson circulation

L11-12: increase of UV in the tropics or mid latitudes?

Increase of UV level is expected in the tropics. We rewrote this part as follows:

… tropics, which causes a decrease in the mid-latitudes and increase in the tropics in surface level of UV radiation…

L20: element? You need a better word here

We rewrote this sentence as follows:

The stratospheric ozone layer plays an essential role in…

L31: The following studies should also be mentioned here

https://doi.org/10.5194/acp-18-7557-2018

https://doi.org/10.1038/s41612-018-0052-6

Done.

L38: GHG was first mentioned in L23

The abbreviation was moved to the line 23.

L84: space after full stop

Done.

L89: decrease (Keeble et al., 2021).

Done.

L102: what is "slightly" comparable;?

Slightly compatible means that simulated trends show some signs of agreements but generally far from the observations. We reformulated this as follows:

"..are not completely consistent…"

L115: Hue et al. (2016)

Done.

L119: CCMI campaigns? Sounds a filed campaign, not modelling experiments

Agree. However, we excluded the paragraph with this line from the revised paper.

L168: respectively (Zhao et al., 2020)

Done.

L192: indices

Done.

L241, 245: delete content, use amount or concentration instead

Done.  "Content" word was exchanged with "concentration".

L248: signs of increase

Done.

L251-252: how NOx produces ozone in the lower stratosphere?

The increased NOx might still contribute to ozone production in the lower stratosphere via smog reactions (e.g., Wang et al., 1998). It was added to the paper's text.

References:

Wang, Y., Jacob, D. J., and Logan, J. A.: Global simulation of tropospheric O3-NOx-hydrocarbon chemistry: 3. Origin of tropospheric ozone and effects of nonmethane hydrocarbons, , 103, 10,757–10,767, https://doi.org/10.1029/98JD00156, 1998.

L253: do not start a sent with AND

Agree. We have merged these two sentences.

L260: These increases of 0.13 and 0.27 DU are significant?

Yes, they are statistically significant.

L269: Can you please give another reference for this. It is known long before, not in 2021

Agree. We have exchanged the given citation with those below:

Thompson, B. A., P. Hartwick, and R. R. Reeves Jr. (1963), Ultraviolet absorption coefficients of CO2, CO, O2, H2O, N2O, NH3, NO, SO2, and CH4 between 1850 and 4000 A, J. Geophys. Res., 68, 6431–6436.

Solomon, S., Garcia, R. R., Olivero, J. J., Bevilacqua, R. M., Schwartz, P. R., Clancy, R. T., and Muhleman, D. O.: Photochemistry and transport of carbon monoxide in the middle atmosphere, Journal of Atmospheric Sciences, 42, 1072–1083, https://doi.org/10.1175/1520- 0469(1985)042<1072:PATOCM>2.0.CO;2, 1985.

L299: as expected

Done.

L369: delete the "expected"

Done.

L382: it barely changes? Then you write a change of -4DU/decade? Delete "barely"

Done.